# Chemical and Physical Characterization of Sorghum Milling Fractions and Sorghum Whole Meal Flours Obtained via Stone or Roller Milling

**DOI:** 10.3390/foods10040870

**Published:** 2021-04-16

**Authors:** Rubina Rumler, Denisse Bender, Sofia Speranza, Johannes Frauenlob, Lydia Gamper, Joost Hoek, Henry Jäger, Regine Schönlechner

**Affiliations:** 1Department of Food Science and Technology, University of Natural Resources and Life Sciences, Muthgasse 18, 1190 Vienna, Austria; rubina.rumler@boku.ac.at (R.R.); denisse.bender@boku.ac.at (D.B.); sofia.speranza@boku.ac.at (S.S.); lydia.gamper@students.boku.ac.at (L.G.); henry.jaeger@boku.ac.at (H.J.); 2Hans Frauenlob Hochmühle, Dorf 15, 5325 Plainfeld, Austria; johannes@frauenlob.at; 3Department of Food Process Engineering, Wageningen University and Research, 6708 PB Wageningen, The Netherlands; joost-hoek@hotmail.com

**Keywords:** sorghum, milling fractions, whole-grain, chemical composition, physical properties

## Abstract

Due to climate change sorghum might gain widespread in the Western countries, as the grain is adapted to hot climate. Additionally sorghum contains a notable amount of health-promoting nutrients. However, Western countries do not have a long history of sorghum consumption, and thus little experience in processing it. Milling systems in these areas were mostly developed for wheat or rye milling. In the present work, the effectiveness of sorghum milling when using a stone and a roller milling system (pilot scale) was investigated as well as its impact on the chemical and physical properties of the obtained flour fractions and whole-grain flours. Results showed that both milling systems could be successfully adapted to producing chemically and physically distinct flour and bran fractions from the small sorghum kernels. Fractions with increased bran material that contained higher amounts of ash, protein, fat, total dietary fiber, and total phenolic content but less starch, showed enhanced water absorption indices and water solubility indices. Interestingly, no significant difference was found in the ash and fat content of the different fractions obtained from stone milling. Overall, the study provided information on the production and composition of distinct flour fractions, which offer a wider range of future food applications.

## 1. Introduction

Sorghum is the fifth most produced cereal worldwide and it is mostly harvested in Africa, followed by America and Asia [1]. Traditionally, sorghum has been used in Africa for foods such as porridges, beers, or flat breads. Outside of Africa, sorghum is mainly used as animal feed or for bioethanol production [2]. However, the cultivation of sorghum may also be beneficial for places like Europe because sorghum exhibits high tolerance to hot climate and genetic diversity. Moreover, it is rich in constituents with presumed health benefits, such as resistant starch, dietary fiber, unsaturated fatty acids, micronutrients, and polyphenols [3]. Numerous in vitro and in vivo studies have indicated that sorghum polyphenols may play a role in cancer prevention, improve glycemic response and inhibit inflammation [4]. Overall, sorghum may be considered a “climate-smart” cereal with a balanced nutritional profile.

When using cereals for food processing, flour production is often the first step. As flour properties and composition greatly affect end-product quality, milling, and fractionation processes play a crucial role. Knowledge on sorghum processing has been mostly gained in Africa and Asia. Some studies have described grinding of sorghum (whole-grain) flour using lab scale mills or local milling systems, e.g., hammer mills [5,6]. Moreover, fractionation by sieving was already performed in the past for several reasons such as the removal of fractions rich in phytic acid [7]; although, others claimed a better nutritional profile when sieving was not applied [8]. Sorghum flour sieving to different particle sizes might be helpful to improve the quality of particular food. Biscuits produced from flours with defined particle sizes of 180 and 251 µm had a better consumer acceptance than those produced from flours with smaller particle sizes (below 180 µm) [6]. In contrast, gluten-free bread produced from 60% sorghum extraction flour showed softer texture and higher specific volume than gluten-free bread from whole-grain sorghum flour [9]. Hence, sieving might be a necessary processing step when using sorghum for human food. However, transferring research from traditional and lab scale mills, where fractionation is typically obtained by sieving whole-grain flour, to an industrial scale (wheat) milling system with numerous passages is challenging. Few studies have assessed if the difference in kernel morphology between sorghum and wheat or rye allow for it to be processed via milling systems designed for wheat or rye. Only in 2018, a study was published on sorghum fractionation using a pilot scale milling process (Hal Ross flour mill Kansas State University, Manhattan, NY, USA) that consisted of several break and reduction passages, as well as purification passages. Sorghum flour was successfully separated into different passages, after which the obtained fractions were recombined to a flour, a middling and a germ fraction. However, these flour fractions were not chemically analyzed [10]. Information on the chemical composition of single fraction flours is not only a useful tool to estimate the effectiveness of milling and fractionation, but also to obtain comprehensive information on the distribution of nutrients within the sorghum kernel.

In Austria, different varieties of sorghum are cultivated in very small quantities. The harvested grain is used for animal feed and bioethanol production, but not for human nutrition. Therefore, these sorghum varieties have neither been investigated for their food processability nor been chemically analyzed before. In order to assess the potential of these Austrian grown sorghum varieties, the aim of this study was (1) to produce sorghum whole-grain flours, as well as flour fractions; and (2) to characterize all the obtained flours and fractions for their chemical and physical properties. Milling and fractionation of sorghum was carried out using two pilot scale milling systems (stone mill and roller mill). For comparison purposes, an industrial scale whole-grain flour, milled by a roller milling system was included. Chemical and physical analyses were performed to evaluate on the other hand, whether the (wheat) pilot scale stone mill and/or the roller milling system can be used for sorghum milling and fractionation. For roller milling, all single fractions were collected separately, in order to follow the performance and effectiveness of this rather complex milling system (see Figure 1). The study additionally aimed to assess the chemical and physical properties of the different flour fractions; in order to provide information on their potential future food use. To incorporate sorghum flour in the Western food industry, special flour qualities for selected food groups may be required. Therefore, local milling technologies have to be tested for suitability to produce appropriate flours from sorghum grains. This research sheds light especially on roller milling, as to date this milling methodology has not yet been evaluated for sorghum milling.

## 2. Materials and Methods

### 2.1. Material

Dried and dehusked red sorghum grains (var. *Armorik*), where grown on an experimental farm in Hörsching, Austria and harvested in 2019. They were obtained from Caj. Strobl Naturmühle GmbH (Linz-Ebelsberg, Austria) and were used for the milling trials (Section 2.2). For comparison a whole-grain sorghum flour from the same variety and milled commercially (industry scale roller milling system) by Strobl Naturmühle GmbH (Linz-Ebelsberg, Austria) was included in the analytical experiments.

### 2.2. Milling of Sorghum Grain

Sorghum grains were milled by two different milling systems. From both of these milling systems a whole sorghum flour (WSF) was obtained.

DFS-PSM (dry flake squeezer—pilot scale stone mill): Several pre-trials were performed to select the appropriate equipment and settings for stone milling. Based on these results (data not shown) a pilot scale dry flake squeezer (Goldflocke GF-40-Super, M&A Hommel GmbH, Wülfrath, Germany) and a pilot scale stone mill (ABC Hansen Universal Mill, ABC Hansen, Randers, Denmark) produced a WSF, which was separated by a sifter (Einkasten-Plansichter, Rüter Maschinen, Hille, Germany) into three flour fractions with particle sizes >400 µm, 400–180 µm, and <180 µm. The purpose of squeezing the kernels prior to stone milling is to increase the flour yield. As a first step, the sorghum kernels were squeezed in the dry flake squeezer at 1000 kg/h, then they were ground in the pilot scale stone mill (PSM), set with the smallest possible gap between the stones, 507 revolutions per minute and an hourly rate of 100 kg/h. The gap between the stones was estimated 0.2–0.5 mm, which could not be quantified more accurately. One stone of the mill veered and the other one backed. For fractionation, the whole-grain sorghum flour was first sieved with a 400 µm sieve at 2000 kg/h. The <400 µm fraction was further sieved through a 180 µm sieve, thus obtaining two further fractions (400–180 µm and <180 µm). DFS-PSM was performed from 100 kg of sorghum grain.

PRM (pilot scale roller mill): The second milling trial applied a pilot scale roller milling plant made of several roller mills (B1: Phenomill Walzenstuhl, Rückert Mühlen und Anlagentechnik GmbH & Co. KG, Altdorf bei Landshut, Germany; B2, B3, B4, B5, C1, C2, C3, C4, R1, R2, and R3: Gliederwalzenstuhl, Happle Maschinenfabrik Anlagenbau Gmbh, Weißenhorn, Germany; C5:C6: Einfachwalzenstuhl, A. Nuß & Vogl, Krems, Austria) and plansifters (Schubladenplansifter, M.Hohl, Graz, Steiermark) after each passage. Milling was performed with an hourly rate of 150 kg/h. The configuration and settings of this pilot scale roller milling plant were established according to pre-trials on Bühler MLU-202 (Fa. Bühler, Uzwil, Switzerland), which is a lab scale roller milling system (results not presented here). The process of (pilot scale) roller milling/sifting is based on various breaking and reduction steps (see Figure 1). Usually four fractions, namely, flour, fine bran, coarse bran, and residue flour (often claimed as animal feed flour) are obtained when using this milling system. In this study all single passage flour fractions were separately collected into overall 14 samples. B1–B3 flour fractions were obtained from the breaking steps, while C1–C4 flour fractions were obtained from the reduction steps. C5/C6/B4/B5 were combined to one “dark” flour fraction in this study. R1–R3 fractions were obtained from the semolina milling steps (semolina milling is not always performed commercially). Coarse and fine bran fractions were obtained after all B and C milling passages. The breaking system is mainly responsible for separating endosperm from bran. With each breaking passage, the respective rollers show a smaller gap [11]. Middlings are a byproduct of every break passage (B). Each passage is followed by a plansifter, which is responsible for separating endosperm flour, bran, and middlings from each other. These middling fractions were introduced into the reduction system (C), which is constructed to leach out the endosperm flour from middlings [11]. The very last coarse flour fraction after the reduction system (after C6) was collected as animal feed fraction. Roller milling plants do typically not provide a WSF. This can be obtained by re-mixing the flour fractions in the corresponding yield ratio, which is often called reconstituted whole-grain flour. PRM was performed from 400 kg of sorghum grain.

All produced WSFs, as well as all defined flour fractions were analyzed, as described below. From the 14 flour fractions obtained in the second milling system five fractions were selected, representing different sections of the sorghum kernel: bran fine (outer layer), B1, C1 (white endosperm flour), C4, C5/C6/B4/B5 (dark endosperm flour), as well as a reconstituted WSF. The selected fractions were chosen in order to follow if fractions from different parts of the kernel were characterized by different chemical or physical properties.

### 2.3. Color Analyses

Color analyses were carried out as described in Bender et al. [12]. A DigiEye device (VeriVide, Leicester, United Kingdom) including a D-90 Nikon camera (Tokyo, Japan) was used to characterize the flour color. Results were expressed by L*a*b-values (CIELAB parameters), and the measurements were carried out in triplicate.

### 2.4. Chemical Analyses

Moisture content was determined by ICC standard method Nr. 110/1 [13]. Ash content was analyzed according to ICC standard method Nr. 104/1 [14] by pre-ashing the samples in a quick incinerator (Harry Gestigkeit GmbH SVR/E, Germany) until no smoke was no longer detectable, then ashed in a muffle furnace (Carbolite Elf 11/6B, England) at 600 °C for six hours. ICC Standard method Nr. 105/2 [15] was used to determine crude protein contents by Kjeldahl (Büchi KjelDigester K-449, Fawil, Switzerland) with a nitrogen conversion factor of 6.25. Fat analyses were carried out based on ICC method Nr. 136 [16], in which 6 g samples were extracted with petroleum ether using a Soxhlet apparatus (Foodalyt RS 60, Bremen, Germany). Total starch and total dietary fiber were determined by using enzymatic test kits from Megazyme (Megazyme International Ireland Ltd., Wicklow, Ireland) following the standard methods AACC No. 76-13.01 [17] and AACC No. 32-07 [18], respectively. Ash, protein, fat, starch, and dietary fiber were reported as percentage in dry matter. Total polyphenols (TPC) were extracted and determined according to Siebenhandl et al. [19] with minor modifications. In brief, 0.6 g flour samples were mixed well with 12 mL of 2 M sodium hydroxide in a 50 mL centrifuge tube. Afterwards, tubes were shaken under nitrogen atmosphere in a dark room. After 16 h, the mixtures were brought to pH 3.5 with 3 M citric acid and centrifuged 5 min at 3092× *g*. Supernatants were separated from the sediment and the liquid phase was collected. After adding 10 mL ethyl acetate to both, the solid and the liquid phases, the mixtures were shaken for 5 min and afterwards centrifuged for 2 min at 3092× *g*. For these analyses ethyl acetate was chosen as extraction aid in order to extract the liberated phenolics, as well as the previously unesterified phenolics, from the aqueous extract. Ethyl-acetate is a suitable solvent for the phenolics and, in contrast to more polar solvents like methanol or ethanol, can be separated from the aqueous phase via centrifugation. The supernatants (ethyl acetate phases) of the solid and the liquid phases were collected together in a shaded round bottom flask. The extraction with ethyl acetate was repeated twice. Samples were concentrated using a rotavapor (R-300—Büchi Labortechnik, Switzerland) and dissolved in 4 mL of 50% methanol. For TPC determination 100 µL of the extracts were mixed into 0.5 mL of freshly prepared 10-fold diluted Folin-Ciocalteu reagent. After 2 min waiting time 0.8 mL sodium carbonate (75 g/L) was added. The mixtures were incubated for 5 min at 50 °C. The absorbances were read at 755 nm against a blank. The results were expressed as mg ferulic acid equivalent (FAE) per 100 g dry substance. All chemical analyses were carried out in triplicate.

### 2.5. Water Absorption Index and Water Solubility Index

The water absorption index (WAI) and the water solubility index (WSI) were measured based on the method from Anderson et al. [20]. Briefly, 2.5 g samples were combined with 30 mL distilled water and shaken for 30 min at 30 °C. Afterwards the tubes were centrifuged at 1739× *g* for 10 min and residues were weighed for WAI calculation. Supernatants were collected in aluminum dishes and evaporated for 4 h in a drying cabinet at 103 °C. Dried solids were used for WSI calculation. WAI was expressed as absorbed water in g/g dry sample, WSI was expressed as percentage of dry solids/2.5 g dry sample. Measurements were carried out in triplicate.

### 2.6. Statistical Data Analyses

Statistical data analyses were carried out using Statgraphics Centurion 18, Version 18.1.13 (Statpoint Technologies, Inc., Warrenton, VA, USA). Significant differences were evaluated by performing an ANOVA and Fisher’s least significant test with a significance level of α = 0.05. Results were expressed as mean ± standard deviation and significant differences between results were expressed by different superscript letters.

## 3. Results and Discussion

### 3.1. Milling

Milling yields obtained from the two investigated milling systems are visualized in Table 1.

The most abundant flour fraction in DFS-PSM, was the flour with a particle size of <180 µm, 43.9%, followed by almost equal parts of the remaining two fractions >400 µm, 27.5%, and 400–180 µm, 28.6% (see Table 1).

As described in the methodology, all PRM passages were collected separately. As for the PRM milling system, one-fifth of the flour fractions ended in bran material (bran coarse and bran fine) and further 8% in the bran-rich animal feed flour. As expected, the flour fraction B1 was extracted to a very low extent, as it is the first endosperm flour obtained after grain breaking. Higher yields were obtained during the reduction milling system (C1–C5). The flour fraction C1 showed a milling yield of 11%. However, the highest amount within the flour fractions C and B was extracted after the C5/C6 passage, and after the B4/B5 passage, respectively. Total endosperm flour accumulated to 72%, while the remaining 28% were bran and animal feed. Only five flour fractions were selected from this milling system for subsequent chemical and physical analyses and are highlighted in Table 1. With the selection of the fractions it was aimed to investigate different kernel sections (inner and outer endosperm, outer kernel, bran layers), which are derived from different sections of the PRM (Figure 1). Additionally, analyses of the ash content of all 14 obtained flour fractions were carried out (data not shown), which further supported the selection of fractions according to this aim. Whole-grain flour was reconstituted according to the obtained yields of flour that had gone through all passages. However, some of the flour is inevitably lost in a large-scale milling system. In Table 1 the flour fractions are expressed as ratio of all the collected flour (i.e., minus the loss occurring in the passages), which may have caused slight deviations between WSF reconstituted based on fraction yields compared to the original kernel composition. This might have caused an improper WSF reconstitution based on the fraction yields.

Overall, both milling trials were successful to produce not only whole-grain flour, but also different flour fractions. Although the sorghum kernel differ greatly from wheat or rye kernels in size and shape, both the stone mill and the roller milling system could be used and adapted for production of sorghum milling fractions. Endosperm flour was successfully extracted in both cases, as could visually be observed, and will be described by the color and chemical data in the subsequent sections, Well configurated industrial scale mills with even higher number of passages are able to extract up to 80% of endosperm flour in wheat milling.

### 3.2. Color Analyses

In Table 2 it can be seen that the color of the different samples varied widely.

In the DFS-PSM milling fractions lightness (L*-value) increased with decreasing particle sizes. They were also less red and yellow, which indicated a lower content of bran in the finer fractions.

In the PRM milling fractions, the bran fraction had the highest a*-values (red) and b*-values (yellow), as expected. These value trends are further in line with previous findings [9]. However, all other PRM fractions were similar in lightness. While B1 had the lowest b* value, C1 had the lowest a* value. The C4 flour fraction occurred to be more reddish, while C5/C6/B4/B5 was more yellowish, among all B and C flour fractions. Comparing the three WSFs, PRM WSF was the lightest flour, while the other two WSFs did not differ significantly from each other. PRM WSF was reconstituted by mixing fractions according to yield ratio. On the basis of color analyses (a*- and -b*-values), it seemed that less amount of outer layers were included in the IRM WSF (lower a*-, but higher b*-value), than in DFS-PSM and PRM whole-grain sorghum flours. A recent review emphasized the crucial role of color for consumer acceptance of baked goods, and thus sorghum flour of light color may be required for the incorporation into wheat-based bakery products [21]. However, color on the outside of sorghum kernels does not always reflect the kernels’ inner color. By means of the PRM it was feasible to obtain light colored sorghum fractions, although the sorghum kernel was originally dark reddish (see Table 2).

### 3.3. Chemical Analyses

The results of the chemical analyses are displayed in Table 3 and Table 4.

The dry matter content of all analyzed sorghum flours ranged between 88.73 annd 91.72%. Among the sorghum flour fractions obtained from DFS-PSM, no significant differences in ash and fat content were observed. This was also seen in a previous study, which stated no differences in ash, fat, and protein in sieving fractions of whole-grain wheat [22]. The flour fractions of DFS-PSM had a similar ash content but a significantly lower fat content compared to WSF (1.70 ± 0.07% dm and 3.84 ± 0.10% dm, respectively). Protein, starch, TDF (IDF as well as SDF) and polyphenol content of DFS-PSM flour fractions varied among the various fractions, and tended to decreased from the coarser flour fractions to the finer flour fraction, except for starch and TPC. In this case the middle fraction (400–180 µm) displayed a higher content. Interestingly, starch, TDF and TPC content of this middle fraction did not differ from the amount measured in WSF. This leads to the suggestion that the middle fraction had similar proportions of outer and inner grain parts as WSF. Findings of higher protein, TDF and TPC content in outer layer fractions in the present investigation are in agreement with other studies, which indicated the highest protein content in sorghum bran fractions [23,24]. High starch content in high extracted sorghum flours have been reported previously [9,10].

Based on protein, starch, TDF, and total polyphenol content, it seemed that nutritional fractionation within the DFS-PSM sieving took place; although, ash and fat contents did not differ significantly among different fractions. Both, fat and ash, are the predominant nutrients in the germ [25,26]. An explanation for the similar fat and ash composition of the flour fractions is that the germ was not removed as such, but milled into heterogeneous particle sizes, and therefore distributed into all flour fractions, while the other kernel components (seed hulls) remained larger and were thus accumulated in the coarse flour fractions. This occurs due to the fact that in stone milling the kernels are ground more completely between the two milling stones. These are usually set with a very small gap between them (as small as possible, estimated 0.5 mm, see Section 2.2.), which results in more thorough grinding of all kernel parts right away. In roller milling this is different, here the mills of the first passages only break the kernel coarsely in the first passages, allowing for the starchy inner endosperm to be separated from the rather coarse and more “intact” kernel parts. With ongoing roller milling passages the gap between the rolls is decreased.

As described in Section 2.2, in the PRM milling trial all obtained milling fractions were collected separately after each break (B1–B5) or reduction passages (C1–C5), from which the fine bran fraction, B1, C1, and C4 and combined C5/C6/B4/B5 passage flours were selected for chemical analyses. WSF was reconstituted according to the fraction yield ratios after milling. As expected, the bran fraction, which was obtained after several break passages, showed by far the highest amount of ash, fat, TDF (IDF as well as SDF), and total phenolic contents among all PRM flour fractions studied. The present results are in agreement with the publication of Moraes et al. [23]. High presence of phenolic compounds in flours with enriched bran material was also found by Awika et al. [27]. The chemical results of the PRM flour fractions demonstrated that removing bran from sorghum kernels within the PRM trial was successful: Starch rich flour fractions were obtained from the very first break passage (B1), where the sorghum kernel was broken and the first endosperm flour was extracted with a yield of 3% (compare Table 1). The results showed that B1 flour fraction had a very high starch and very low ash, protein, fat, and TDF content. It is difficult to compare these results with the literature, as to our knowledge single sorghum flour fractions obtained from roller milling have never been analyzed before. However, these results seem to suggest that PRM milling worked the way, as it usually works for wheat.

The C1 passage flour fraction showed similar results to the B1 passage fraction flour. On the basis of ash, protein, starch, fat, TDF, and TPC values, it can be concluded that the C1 fraction flour corresponds to a rather pure endosperm flour. In wheat milling, the flour obtained after the C1 passage milling corresponds to a very white endosperm flour, which is categorized as *type W480* according to the Austrian codex Alimentarius [28]. The reduction passage milling (C fractions) is applied to separate the endosperm particles from the middlings fraction that are produced during the break passage milling (B). With increased C passage millings, starch rich endosperm is separated from the middlings, the C4 and C5/C6/B4/B5 flour fractions are thus known to include higher amounts of outer kernel layers and less endosperm parts than the C1 flour fraction. C4 and C5/C6/B4/B5 fraction flours had similar ash, fat, and TDF contents. However, C5/C6/B4/B5 was significantly higher in protein content, lower in starch content and higher in TPC than the C4 fraction. As all analyzed flours fractions showed distinct different chemical composition, it can be concluded that also a roller mill/plansifter milling system seems to be suitable to produce defined milling fractions from sorghum.

As mentioned previously, milling has a great effect on the resulting flour properties. The effect of the two applied milling systems on flour quality can be followed when comparing the whole-grain sorghum flours produced in this study. For comparison, a commercially available whole-grain sorghum flour was included (WSF IRM, produced by an industrial scale roller milling system). Again, it has to be considered that PRM WSF was reconstituted. Ash was the only component which did not differ significantly between the three whole-grain sorghum flours. The highest protein value was observed for PRM WSF, followed by DFS-PSM and IRM. An explanation for the higher protein content in PRM could be an improper reconstitution of WSF based on the fraction yields. Endosperm flour yields may have been underestimated, as an undefined amount of flour residue inevitably remained in the mill. So far, lower protein contents in tartary buckwheat flours obtained from a roller mill compared to a stone mills have been reported [29], but no considerable effect has been found on wheat flour [30]. However, the protein content of the used sorghum variety in this study was overall rather low. Tasie et al. [31] detected a range of 8.2 to 16.5% for protein content in thirty-five Ethiopian sorghum varieties. Starch content of the three sorghum whole-grain flours varied immensely in a range of 48–72% dm. Fat content of PRM WSF and IRM WSF did not differ significantly, but was higher in DFS-PSM WSF. A similar TDF content was found in PRM and IRM, while TDF was lower in PSM, in particular IDF, which was significantly higher in PRM than in DFS-PSM. However, SDF content was not significantly different between the studied WSFs. Similarly to protein, the TPC content was higher in the reconstituted PRM WSF, while the DFS-PSM WSF and IRM WSF were comparable. The variation of nutrients (i.e., starch, protein, and TPC) of PRM WSF might be attributed to the reconstitution process of the whole-grain flour, containing proportionally too much bran.

Generally, rather low and insignificant differences were observed between the stone milled and roller milled flours. Pagani et al. [30] also did not detect considerable chemical differences between stone milled and roller milled flours. Still, and as also emphasized by Pagani et al. [30], popularity of stone milled flour in the population is high, but when looking at the chemical and nutritional results, this is not based on facts and lacks rationality.

### 3.4. WAI and WSI

Results of water absorption index (WAI) and water soluble index (WSI) are given in Table 5.

Flours obtained from DFS-PSM decreased in WAI with decreasing particle size. Al-Rabadi et al. (2012) evaluated comparable results on sorghum flour for similar particle sizes. However, the authors reported no consistent WAI trend in flours of different particle sizes [5]. Cadden [32] reported a higher water binding capacity for higher particle sized bran, suggesting that porosity of material played a role. In our case, it is known that the >400 µm flour fraction had a higher dietary fiber content than the remaining two fractions. High water binding capacities in bran rich flours were shown in the past [33,34]. The same suggestions can be made for the PRM flour fractions. The bran fine flour fraction was by far, the one with the highest water absorption index. The endosperm flour fractions (B1, C1, C4, and C5/C6/B4/B5) showed a water absorption range from 2.47 to 2.75 g/g dm. DFS-PSM and IRM whole-grain flours had similar water absorption levels, while PRM displayed a lower WAI, although its TDF content was higher. This could probably be related to a lower amount of damaged starch, which was not measured in this study, though. Palavecino et al. [35] found a positive correlation between damaged starch and water absorption.

Water solubility indices (WSI) of DFS-PSM flour fractions were lower with decreasing particle size. Looking at the PRM flour fractions, the bran had the highest WSI. Flour fractions from B and C fractionation showed similar WSI ranging from 2.42 to 3.7 1%. Comparing the whole-grain flours, WSI from PRM milling had a significant higher WSI than from PSM or IRM milling. This might also be related to the higher ratio of bran in the reconstituted whole-grain flour of PRM.

Considering the chemical data in combination with the physical properties, it can be stated that the obtained sorghum flour fractions offer different food uses, both, either in terms of intended nutritional enrichment, and in terms of specific processing needs. Some bakery products require very defined physical or rheological conditions of flours, e.g., fine bakery products or wafers. Replacing wheat flour with any other flour therefore has major effects on the end-product quality and requires thorough adaptation of recipe and processing parameters. Detailed knowledge on flour properties is one of most important factors to be considered in this respect. The data obtained in this study for sorghum flour fractions provide a step forward to future use of sorghum and incorporation into Western type bakery products.

Table 2, Table 3, Table 4 and Table 5 visualize various flour fractions different in color, chemical, and physical properties. The possibility of producing those sorghum flour fractions, may provide significant benefits to the food industry. Endosperm flour fractions (e.g., DFS-PSM: <180; PRM: B1 and C1) offer specific baking properties, particularly for, but not restricted to, fine bakery products; also their light color may be advantageous to maintain specific food colors after enrichment, e.g., sandwich toast, noodles, or waffles. Furthermore, low particle size milling fractions (<180 µm) may be useful for gluten-free food development, as lower particle size flours seem to influence product quality (volume and crumb firmness) in a better way than coarse flours [9]. Fractions enriched with polyphenols, protein and fiber, but still possessing moderate water uptake (e.g., DFS-PSM: 180–400 µm; PRM: C4) may be interesting for companies, who intend to promote their products with containing health beneficial potential.

## 4. Conclusions

Results of this research indicated that stone and roller milling, both typically applied for wheat milling, were effective for milling sorghum and could successfully produce defined flour fractions, as well as whole-grain flours.

With regard to stone milling, the chemical and physical properties of the obtained flour fractions were clearly distinct, demonstrating that fractionation was not only based on their particle sizes (>400 µm, 400–180 µm, and <180 µm), but also on their chemical profile.

In the roller milling system, flour fractions were not separated by sieving only, but by a typical sieving/sifting combination after each roller mill passage. Although the size and shape of sorghum kernels differed to wheat, roller milling settings could be adapted to sorghum milling. Chemical and physical properties of defined roller milled flour fractions indicated that sorghum grains were separated into different anatomical parts. Considering the chemical data, it can be concluded that the B1 and C1 flour fractions were highly concentrated from endosperm parts. The C4 and C5/C6/B4/B5 fractions can be defined as dark endosperm flours, which was confirmed by a lower starch content, but a higher ash, protein, fat, and TDF content compared to B1 and C1 flour fractions. As the bran fraction was shown to be largely composed of outer kernel layers and germ, it was confirmed, that it could successfully be separated from the endosperm. The WSF from the roller milling system was significantly higher in protein and TPC, but lower in starch compared to the WSF from stone milling. This was likely the result of an improper reconstitution of WSF based on the fraction yields.

Overall, the results of the chemical and physical analyses of the flour fractions, obtained from stone or roller milling systems, seem to confirm that sorghum milling can be carried out on typical wheat milling systems without detailed or complex technical adaptations. Both milling systems were successful in producing flour fractions with distinct chemical and physical properties. The two milling systems differed in their ability to produce distinct fractions, with roller milling being superior to stone milling. Depending on the intended food use, the availability of defined flour fractions can be advantageous for selected food applications [6,9] as some bakery products require very defined physical or rheological conditions of flours. Knowing the properties of flour is thus one of the most important factors. However, it should be considered that using flour fractions is not always economically and nutritionally feasible compared to the use of whole-grain flours.

This study provided information on defined sorghum flours and flour fractions that might facilitate the future use of sorghum in the Western diet.

## Figures and Tables

**Figure 1 foods-10-00870-f001:**
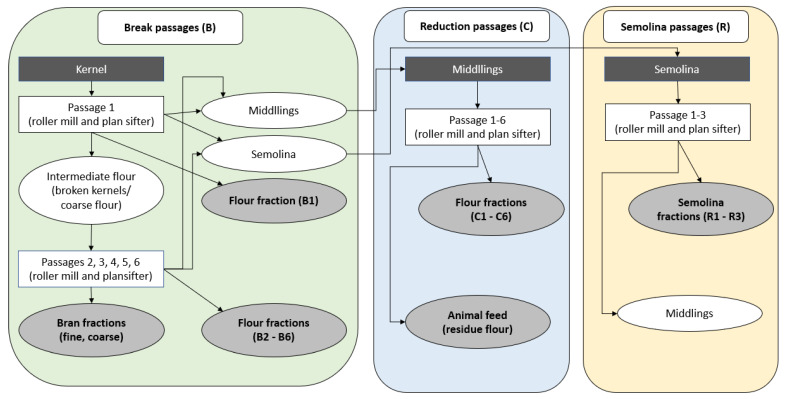
Milling scheme for the roller milling system (PRM). All extracted flour fractions (B1–B6, C1–C6, R1–R3, fine, and coarse bran fraction as well as animal feed) were collected separately.

**Table 1 foods-10-00870-t001:** Milling yields (%) obtained from DFS-PSM and PRM milling systems.

Sample	Yield (% of Whole-Grain Flour)	
DFS-PSM		
>400 µm *	27.5	
400–180 µm *	28.6	
<180 µm *	43.9	
PRM		
Bran coarse	18	28%
Animal feed	8	
Bran fine *	2	
B1 *	3	72%
B2	6	
B3	1	
C1 *	11	
C2	3	
C3	14	
C4 *	6	
C5/C6/B4/B5 (combined collection) *	24	
R1	3	
R2 R3	0 1	

* Samples, which were chemically and physically analyzed for the present study. DFS-PSM: Dry flake squeezer—pilot scale stone mill, PRM: Pilot scale roller mill.

**Table 2 foods-10-00870-t002:** Color analyses of sorghum kernels, flour fractions and whole-grain sorghum flours (WSFs) obtained from different milling systems (DFS-PSM, PRM, IRM).

Sample		L*	a*	b*
Kernels	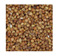	44.4 ± 0.6	18.49 ± 0.49	24.80 ± 0.53
DFS-PSM				
WSF	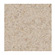	73.5 ± 1.8 ^b;A^	6.50 ± 0.73 ^b;B^	12.22 ± 0.46 ^b;B^
>400 µm	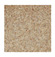	62.7 ± 2.3 ^a^	9.42 ± 0.59 ^c^	18.86 ± 0.37 ^c^
400–180 µm	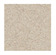	79.1 ± 0.4 ^c^	5.64 ± 0.41 ^ab^	12.46 ± 0.08 ^b^
<180 µm	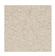	82.6 ± 0.5 ^d^	4.84 ± 0.39 ^a^	11.53 ± 0.41 ^a^
PRM				
WSF	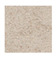	79.2 ± 0.5 ^b;B^	6.10 ± 0.06 ^bc;AB^	11.04 ± 0.37 ^d;A^
Bran fine	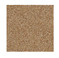	53.6 ± 0.8 ^a^	12.63 ± 0.06 ^d^	20.65 ± 0.08^e^
B1	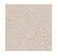	83.8 ± 0.3 ^c^	5.88 ± 0.00 ^b^	8.05 ± 0.01 ^a^
C1	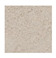	81.3 ± 3.5 ^bc^	5.23 ± 0.44 ^a^	10.48 ± 0.20 ^c^
C4	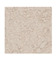	80.7 ± 0.9 ^b^	6.28 ± 0.01 ^c^	10.04 ± 0.03 ^b^
C5/C6/B4/B5	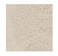	80.9 ± 1.9 ^bc^	5.34 ± 0.06 ^a^	11.36 ± 0.35 ^d^
IRM				
WSF	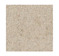	74.9 ± 0.8 ^A^	5.46 ± 0.46 ^A^	13.11 ± 0.27 ^C^

Means (n = 3 ± standard deviation) with same superscript lowercase letters within a column and within the same milling system are not significantly different at *p* > 0.05 (Fisher’s Least Significant Difference test). Means (n = 3 ± standard deviation) with same superscript capital letters (comparison of the WSFs) are not significantly different at *p* > 0.05 (Fisher’s Least Significant Difference test). DFS-PSM: Dry flake squeezer—pilot scale stone mill, PRM: Pilot scale roller mill, IRM: industry scale roller mill.

**Table 3 foods-10-00870-t003:** Dry matter (%), ash (% dm), protein (% dm), starch (% dm), and fat content (% dm) of flours obtained from the three different milling systems.

Sample	Dry matter (%)	Ash (% dm)	Protein (% dm)	Starch (% dm)	Fat (% dm)
DFS-PSM					
WSF	90.40 ± 0.04 ^c;B^	1.70 ± 0.07 ^a;A^	7.74 ± 0.06 ^b;B^	71.90 ± 2.53 ^c;C^	3.84 ± 0.10 ^b;B^
>400 µm	89.82 ± 0.01 ^b^	1.70 ± 0.22 ^a^	9.31 ± 0.13 ^d^	56.38 ± 2.20 ^a^	3.01 ± 0.07 ^a^
400–180 µm	88.73 ± 0.13 ^a^	1.61 ± 0.01 ^a^	8.58 ± 0.03 ^c^	70.73 ± 1.16 ^c^	2.98 ± 0.01 ^a^
<180 µm	88.82 ± 0.01 ^a^	1.67 ± 0.06 ^a^	7.48 ± 0.01 ^a^	66.17 ± 3.059 ^b^	2.94 ± 0.05 ^a^
PRM					
WSF	89.67 ± 0.07 ^a;A^	1.75 ± 0.02 ^b;A^	8.78 ± 0.06 ^e;C^	48.46 ± 2.46 ^c;A^	3.48 ± 0.02 ^b;A^
Bran fine	90.98 ± 0.84 ^ab^	3.80 ± 0.19 ^c^	11.93 ± 0.19 ^f^	6.78 ± 1.24 ^a^	5.12 ± 0.32 ^c^
B1	91.33 ± 1.23 ^b^	1.26 ± 0.10 ^a^	4.86 ± 0.01 ^a^	57.22 ± 0.15 ^e^	2.25 ± 0.03 ^a^
C1	90.77 ± 0.70 ^ab^	1.16 ± 0.14 ^a^	6.79 ± 0.05 ^b^	60.32 ± 0.70 ^f^	2.27 ± 0.04 ^a^
C4	91.72 ± 1.27 ^b^	1.70 ± 0.08 ^b^	7.12 ± 0.03 ^c^	53.18 ± 0.87 ^d^	3.60 ± 0.06 ^b^
C5/C6/B4/B5	91.69 ± 0.50 ^b^	1.67 ± 0.04 ^b^	8.34 ± 0.02 ^d^	43.72 ± 2.29 ^b^	3.43 ± 0.34 ^b^
IRM					
WSF	90.39 ± 0.15 ^B^	1.72 ± 0.02 ^A^	7.39 ± 0.03 ^A^	56.29 ± 0.70 ^B^	3.55 ± 0.04 ^A^

Means (n = 3 ± standard deviation) with same superscript lowercase letters within a column and within the same milling system are not significantly different at *p* > 0.05 (Fisher’s Least Significant Difference test). Means (n = 3 ± standard deviation) with same superscript capital letters (comparison of the WSFs) are not significantly different at *p* > 0.05 (Fisher’s Least Significant Difference test). DFS-PSM: Dry flake squeezer—pilot scale stone mill, PRM: Pilot scale roller mill, IRM: industry scale roller mill.

**Table 4 foods-10-00870-t004:** Total phenolic content (TPC), soluble dietary fiber (SDF), insoluble dietary fiber (IDF), and total dietary fiber (TDF) in dry matter of flours obtained from the three different milling systems.

Sample	IDF (% dm)	SDF (% dm)	TDF (% dm)	TPC (mg FAE/100g dm)
DFS-PSM				
WSF	7.03 ± 0.09 ^b;A^	0.62 ± 0.29 ^ab;A^	7.65 ± 0.21 ^b;A^	152.2 ± 9.2 ^b;A^
>400 µm	9.74 ± 0.24 ^c^	0.27 ± 0.35 ^a^	10.01 ± 0.57 ^d^	317.4 ± 14.5 ^c^
400–180 µm	7.65 ± 0.27 ^b^	0.88 ± 0.01 ^bc^	8.53 ± 0.34 ^c^	161.8 ± 8.9 ^b^
<180 µm	5.22 ± 0.75 ^a^	1.11 ± 0.19 ^c^	6.33 ± 0.59 ^a^	122.3 ± 6.8 ^a^
PRM				
WSF	7.92 ± 0.02 ^c;B^	0.89 ± 0.14 ^c;A^	8.82 ± 0.12 ^c;B^	237.2 ± 12.0 ^d;B^
Bran fine	50.23 ± 0.82 ^d^	1.87 ± 0.23 ^d^	52.1 ± 0.59 ^d^	1131.5 ± 36.9 ^f^
B1	1.76 ± 0.09 ^a^	0.51 ± 0.16 ^ab^	2.27 ± 0.25 ^a^	76.5 ± 3.3 ^b^
C1	2.39 ± 0.03 ^ab^	0.30 ± 0.09 ^a^	2.69 ± 0.12 ^a^	37.4 ± 4.7 ^a^
C4	2.64 ± 0.33 ^b^	0.80 ± 0.20 ^bc^	3.43 ± 0.49 ^b^	54.7 ± 6.2 ^ab^
C5/C6/B4/B5	3.10 ± 0.21 ^b^	0.92 ± 0.08 ^c^	4.02 ± 0.21 ^b^	123.2 ± 6.5 ^c^
IRM				
WSF	7.77 ± 0.75 ^AB^	0.98 ± 0.11 ^A^	8.76 ± 0.73 ^B^	155.7 ± 9.5 ^A^

Means (n = 3 ± standard deviation) with same superscript lowercase letters within a column and within the same milling system are not significantly different at *p* > 0.05 (Fisher’s Least Significant Difference test). Means (n = 3 ± standard deviation) with same superscript capital letters (comparison of the WSFs) are not significantly different at *p* > 0.05 (Fisher’s Least Significant Difference test). DFS-PSM: Dry flake squeezer—pilot scale stone mill, PRM: Pilot scale roller mill, IRM: industry scale roller mill.

**Table 5 foods-10-00870-t005:** Water absorption index (WAI) and water soluble index (WSI) of flours obtained from the three different milling systems.

Sample	WAI (g/g)	WSI (%)
DFS-PSM		
WSF	2.67 ± 0.02 ^d;B^	3.47 ± 0.22 ^a;A^
>400 µm	2.63 ± 0.03 ^c^	9.03 ± 0.37 ^c^
400–180 µm	2.33 ± 0.02 ^b^	7.38 ± 0.73 ^b^
<180 µm	2.29 ± 0.02 ^a^	6.26 ± 1.48 ^b^
PRM		
WSF	2.33 ± 0.06 ^a;A^	6.88 ± 0.78 ^b;B^
Bran fine	4.46 ± 0.16 ^d^	9.70 ± 2.05 ^c^
B1	2.54 ± 0.03 ^b^	2.42 ± 0.10 ^a^
C1	2.75 ± 0.01 ^c^	2.84 ± 0.06 ^a^
C4	2.47 ± 0.22 ^ab^	3.18 ± 0.06 ^a^
C5/C6/B4/B5	2.64 ± 0.02 ^bc^	3.70 ± 0.41 ^a^
IRM		
WSF	2.62 ± 0.10 ^B^	3.96 ± 0.12 ^A^

Means (n = 3 ± standard deviation) with same superscript lowercase letters within a column and within the same milling system are not significantly different at *p* > 0.05 (Fisher’s Least Significant Difference test). Means (n = 3 ± standard deviation) with same superscript capital letters (comparison of the WSFs) are not significantly different at *p* > 0.05 (Fisher’s Least Significant Difference test). DFS-PSM: Dry flake squeezer—pilot scale stone mill, PRM: Pilot scale roller mill, IRM: industry scale roller mill.

## Data Availability

All data are included within the article.

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
