# Peer review of "Chemical and Physical Characterization of Sorghum Milling Fractions and Sorghum Whole Meal Flours Obtained via Stone or Roller Milling"

_foods, 2021, doi:10.3390/foods10040870_

Round 1
Reviewer 1 Report
Review Foods 1161609
Title: Chemical and Physical Characterization of Sorghum Flour Fractions and Sorghum Whole-grain Flours obtained from different Milling Systems
L 59-62 “In 2018, 59 a study was published on sorghum fractionation using a pilot scale milling process (Hal 60 Ross flour mill Kansas State University, Manhattan, US) that consisted of several break and 61 reduction passages, as well as purification passages.”
The above sentence is not adding any value to the introduction. I suggest to erase it. The authors should tighten their Introduction and be more direct in their description in the sentences below the above mentioned sentence.
L 84-89 Bread the long sentence into at least two sentences.
L 84 “grown on an experimental farm” use “grown in …”
L 86 “trials (chapter 2.1). For comparison a” There are sections and not chapters – please correct
L 97-98 “were applied to produce ” erase “were applied to produce” and add a “d” to the verb to read “produced”
L 147 “the flour colors.” Erase the “s”
Throughout the Materials and methods do not use all caps for the equipment, use just the first letter cap and the rest small caps
L 204-206 “The most abundant flour fraction in DFS-PSM, was the flour with a particle size of 400 µm and 400-180 µm).” Please give the percentages of each size.
L 274 “compared to WSF (1.7 ± 0,057% dm and 3,84 ± 0,103% dm, respectively.” Erase commas and use periods
L 278-289 “Interestingly, starch, TDF and TPC content of this middle fraction did not differ from the amount measured in WSF.” The authors should try to explain this finding
“However, not enough data is available on the chemical distribution of nutrients within a sorghum kernel.” Even if there is not enough publish data the authors should try to explain their findings
L 284-285 “it can be concluded that the coarse (>400 µm) flour fraction contained a substantial amount of outer kernel layers.” This section and statement is over-simplified. The authors did not need to carry out the experiment to arrive to such statement; it is common sense. What is expected is that the authors will manage differences in percentage of change. This is a major weakness of this work.
L 298 “ground more intensely or completely” erase the indicate in red font and strikethrough
L 299 “set with a very small gap between them,” Give the gap distance
L 306 “flours_were” erase the underscore between flours and were
L 314 “sorghum kernels within the PRM trial was successful.” The authors statement should be as much quantitative as possible.
L 316 “the first endosperm flour was successfully extracted.” We are not interested in qualitative descriptions; efforts should be made to be more quantitative throughout the manuscript.
Fig. 1 – capitalize every heading and fraction in the graphic
Table 1. The last column needs a heading – it is orphan. It could be “Cumulative Yield”
Table 2. Report L* and values with tenths (one decimal) and a* and b* with hundredths (two decimals) for both the values and standard deviations. Correct when citing these color values in the narrative. Recommendation Lfor the footnote –
Erase this sentence: Results are expressed as mean ± standard deviation (n=3).
Values with same superscript lowercase letters within a column and within the same milling system do not differ significantly at p > 0.05.
And replace to read:
Means (n=3 ± standard deviation) with the same superscript lowercase letters within a column and within the same milling system are not significantly different at P < 0.05.
The name of the test for mean differences should be mentioned in parenthesis after the P < 0.05.
Do the same with the next statement.
Means (n=3 ± standard deviation) with the same superscript capital letters (comparison of the WSFs) and within the same row are not significantly different at P < 0.05.
Do the suggested corrections in all the tables.
Table 3. change the heading to read “Dry matter (%)”
Correct such that all the values in the columns have the same number of decimal places.
Table 4. TP values should only be reported with one decimal place, the same the standard deviations.
Table 5. Standard deviations should be reported only with hundredths (two decimal palces)
Author Response
see our comments in the attached word file

Reviewer 2 Report
Journal: Foods; Manuscript ID: foods-1161609; Type of manuscript: Article
The Authors investigated the potential use of traditional wheat milling systems for sorghum milling, and they characterized, from a chemical and physical point of view, the various milling fractions obtained from the different milling systems. Despite the purposes, the development of the Manuscript needs a strong revision to reach a higher scientific level. In general, English needs a deep revision throughout the manuscript. All the sections, moreover, have some lacking points. Introduction, for instance, should include some more information on sorghum nutritional benefits and on why it is so important to our days (thus justifying the importance of making efforts focused on its technological exploitation, milling included). M&M should include more explanations of the selections performed, and should include the evaluation of the starch damage level of the various fractions. This parameter, in fact, can give a lot of fundamental information on the impact of the milling steps on the end-products. R&D need a deep revision, too: most of this section is focused on repeating data reported in the Table, while very few attempts have been made by the Authors to discuss them (via commenting, evidencing, relating their various findings, etc,). Conclusions as well are a mixture of R&D, Aim, M&M. Here below, more specific comments have been reported.
Abstract
- 16: “...climate. Additionally, ...”
- 21-23: This is quite obvious, isn’t it?
Introduction
- 39: This part should be enlarged, highlighting the health-potential benefits of sorghum or at least its nutritive value.
- 35-42: This part should be enlarged, in general, to give more evidence and importance to this smart cereal.
- 49-50: “... a better nutritional profile when sieving was not applied”.
- 53-55: please, explain these findings in a better way.
- 58-59: please, specify sorghum and wheat particle sizes.
- 63: ... separated by means of different steps, after which ... (?)
- 67: ... but also to evaluate the chemical composition of different parts of sorghum kernels.
- 72: ... the aim of this study ...
- 73: ... all the obtained ...
- 74: ... for their chemical and physical properties.
- 77-79: ... and physical analyses were performed to evaluate ... as well as to give preliminary insights into the chemical composition of different zones of sorghum kernels.
- 79-81: This part should be better explained or deleted.
Materials and Methods
- 86: ... (chapter 2.1) ... is there a mistake here?
- 98: ... were used ...
- 95-137: In one case, a squeezing step was adopted; this preliminary step was absent in the second milling system investigated. Why? How many times were the two processes replicated? Which is their CV%?
- 103-104: ... set with the smallest possible gap between the stones (=...), and operating at 507 rpm, 100kg/h.
- 104: please, delete the sentence “One stone of the mill veered and the other one backed”.
- 106: ... please, delete “From the two fractions...”. The <400mm fraction was further sieved through a 180mm sieve, thus obtaining two further fractions (...).
- 109: ...milling plant made of several ...
- 115: ... is based on various breaking and reductions steps (Fig. 1).
Fig, 1: this figure needs a new screenshot since a couple of automatic grammar corrections are highlighted in red!
- 117; ... are obtained when using this milling system.
- 119: ... were obtained from the breaking steps, ... were obtained from the reduction steps.
- 121: ... were obtained from the semolina milling steps...
- 125: ... is followed by a plansifter, ...
- 133-136: which is the basis for this selection? Please, add info.
- 146: ... to characterize the color of the flours.
L: 147: ..., and the measurements were carried out in triplicate.
- 160: ... standard methods AACC...
Results and Discussion
In general, English needs a deep revision throughout the whole manuscript, and particularly all along the R&D section. Therefore, not all the English revisions have been reported here below.
- 198: ... obtained from the two investigated milling systems ...
- 207-208: please delete these sentence (they do not add anything new).
- 209: As for the PRM milling system, one-fifth ...
- 212: Higher yields were obtained during the ...
- 215: ... B4/B5 passages, respectively.
- 216-217: Why? Please, add sound motivations for this selection.
- 219: This observation does not correspond to the data reported in Tab.1, from which – apparently – the whole sample batch was collected (100% yield). Please add explanations.
- 223: Please, give evidence for this observation.
- 228: ... that the color of ...
- 228-229: please, delete the sentence related to the sorghum kernels color.
- 238: please, delete the numbers since they are already reported in the Table.
- 241-250: Authors should add comments focused on explaining the obtained results. Here only the results are reported, with no discussion.
- 255: ... from the three...
- 263: ... from the three...
Table 3 and Table 4: mean and SD should be consistent, for the same parameter; additionally, 3 decimals are too much for some analytics. Please, revise and uniform properly.
Table 3: the “minus” symbol is lacking in the SD values.
- 274: Please, check data expression and consistency.
- 276: ... varied among the various fractions, and tended to decrease ...
- 279-280: Actually, this was the purpose of this study. Please, delete or justify.
- 281-285: This is quite obvious and well known. Please, delete or modify.
- 269-289: Again, this is not a discussion consistent with the purposes of the study. Please, revise/modify.
- 290-303: This part needs revision, both for language and discussion.
- 304-335: This part needs revision, both for language and discussion.
- 371: ...from the three ...
- 384-403: This part needs revision, both for language and discussion. The Authors attempts in commenting their findings, in making connections among the various results etc. are very limited; most of the discussion is actually a repletion of the data reported in the Tables.
Conclusions
Conclusions are a mixture of discussion, materials and methods, and aim. They need to be shortened, revised and focused on the effective findings of this study.
Author Response

(The authors gave the same response as above.)

Reviewer 3 Report
manuscript presents the effectiveness of sorghum milling by using stone and a roller milling system on a pilot scale and the impact of these processes on the chemical and physical properties of the obtained flour fractions and whole-grain flours. A commercially available sample (whole grain) was also used to compare and check differences between whole grain flours. The theme is interesting, manuscript is scientific sound and rather well written and presented, however the technological part is trivial and the analytical procedures simple and present routine analysis of flours. The novelty of paper is not well presented concerning current state of the art of similar procedures. Below find also some queries to which authors should consider and respond:
- Abstract: What TDF, TPC, wai and WSI stand for, please mention the whole names when first mentioned.
- Concerning analytical procedures: why the specific fractionations (400μm, 180μm) were chosen? Which are the ICC methods that the authors refer to? these must be also presented at the ref. section. For fat determination did authors performed acid hydrolysis prior to Soxhlet for bound lipids? For TPC why ethyl acetate was chosen to extract polyphenols and not a more polar solvent like methanol or ethanol? Please comment on this.
- Table 1: what are the values of 72% and 28% represent? even though at L. 215-216 are explained at table is not clear
- Table 2: Revise results after checking for significant digits, sometimes results are given as 62.69 ± 2.26 and then 79.11 ± 0.413, why not 79.11 ± 0.41? later on different result expression: 80.7 ± 0.878. Please do the same for results presented at Table 3 and Table 4 and 5. For TPC results one decimal digit is enough, no need for two or three for std deviation.
- Conclusions are supported by results. Which is the best procedure that authors suggest to get better quality flour? the final suggestion is not clear
Minor spelling and grammar mistakes, but as said before i do not find innovation behind the whole manuscript. Major revision and resubmission is my decision.
Author Response

(The authors gave the same response as above.)

Round 2
Reviewer 3 Report
Authors replied to all my comments and performed all changes suggested to improve their manuscript. Hence, now manuscript has been improved significantly.
Author Response
Thank you very much for the additional revision